# Identification of Novel Genes for Cell Fusion during Osteoclast Formation

**DOI:** 10.3390/ijms23126421

**Published:** 2022-06-08

**Authors:** Eunjin Cho, Seongmin Cheon, Mina Ding, Kayeong Lim, Sang-Wook Park, Chungoo Park, Tae-Hoon Lee

**Affiliations:** 1Department of Oral Biochemistry, Dental Science Research Institute, School of Dentistry, Chonnam National University, Gwangju 61186, Korea; ag8414@gmail.com (E.C.); swpark@chonnam.ac.kr (S.-W.P.); 2School of Biological Sciences and Technology, Chonnam National University, Gwangju 61186, Korea; s.cheon1995@gmail.com (S.C.); chungoo.park@gmail.com (C.P.); 3Proteomics Core Facility, Biomedical Research Institute, Seoul National University Hospital, Seoul 03080, Korea; 4Biomedical Sciences Graduate Program, School of Medical, Chonnam National University, Gwangju 61186, Korea; minading1021@gmail.com; 5Center for Genome Engineering, Institute for Basic Science, Daejeon 34126, Korea; limka@snu.ac.kr

**Keywords:** osteoclast, fusion, Calcrl, Marco, Ube3a, CRISPR-Cas9

## Abstract

Osteoclasts are derived from hematopoietic stem cells. Monocyte preosteoclasts obtain resorbing activity via cell–cell fusion to generate multinucleated cells. However, the mechanisms and molecules involved in the fusion process are poorly understood. In this study, we performed RNA sequencing with single nucleated cells (SNCs) and multinucleated cells (MNCs) to identify the fusion-specific genes. The SNCs and MNCs were isolated under the same conditions during osteoclastogenesis with the receptor activator of nuclear factor-κB ligand (RANKL) administration. Based on this analysis, the expression of seven genes was found to be significantly increased in MNCs but decreased in SNCs, compared to that in bone marrow-derived macrophages (BMMs). We then generated knockout macrophage cell lines using a CRISPR-Cas9 genome-editing tool to examine their function during osteoclastogenesis. *Calcrl*-, *Marco*-, or *Ube3a*-deficient cells could not develop multinucleated giant osteoclasts upon RANKL stimulation. However, *Tmem26*-deficient cells fused more efficiently than control cells. Our findings demonstrate that Calcrl, Marco, and Ube3a are novel determinants of osteoclastogenesis, especially with respect to cell fusion, and highlight potential targets for osteoporosis therapy.

## 1. Introduction

Bone remodeling is maintained by the coordinated actions of bone-forming osteoblasts and bone-resorbing osteoclasts [1]. An imbalance between osteoclasts and osteoblasts leads to bone disease, such as Paget’s disease, osteoporosis, osteopetrosis, and rheumatoid arthritis [2,3]. Osteoporosis is a bone-loss disease mediated by active osteoclasts that leads to an increased risk of bone fracture [4]. Pharmacological agents for osteoporosis treatment are used with anabolic agents to increase bone mass or with antiresorptive agents, such as bisphosphonates, to prevent bone resorption [5].

Osteoclasts are derived from hematopoietic stem cells, specifically myeloid precursors [6]. Receptor activators of the nuclear factor-κB ligand (RANKL) and macrophage colony-stimulating factor (M-CSF) are considered major cytokines that initiate osteoclast differentiation [7,8]. RANKL binds to the RANK receptor on the surfaces of osteoclast precursors to activate downstream signaling pathways, including the mitogen-activated protein kinase and Akt pathways, and transcription factors, such as NF-κB, activator protein 1, microphthalmia-associated transcription factor (MITF), c-Fos, and nuclear factor of activated T-cell cytoplasmic-1 (NFATc1) [8,9,10]. These transcription factors promote the expression of osteoclastogenic genes, such as tartrate-resistant acid phosphatase (*TRAP*, *Acp5*), v-ATPase subunit d2 (*Atp6v0d2*), cathepsin K (*Ctsk*), osteoclast-associated receptor (*Oscar*), osteoclast stimulatory transmembrane protein (*OC-STAMP*), and dendritic cell-specific transmembrane protein (*DC-STAMP*) [11].

Cell fusion facilitates the exchange of luminal contents, and it is critical for the proper development of multicellular organs and their functions [12,13,14,15]. Osteoclast multinucleation is a hallmark of maturation and is the latest differentiation step in osteoclastogenesis [15]. During osteoclastogenesis mediated by external stimuli such as RANKL, the podosomes of macrophage-lineage monocytes fuse together to form multinucleated cells (MNCs) [12]. Several molecules, including DC-STAMP and OC-STAMP, are involved in osteoclast multinucleation via cell fusion [15]. In addition, ATP6v0d2 is a vacuolar ATPase that releases protons extracellularly in the resorption lacunae, and multinucleation fails in bone marrow monocytes from *ATP6v0d2*-knockout mice [16]. The expression of these fusogens is regulated by osteoclast factors, such as through NFATc1-mediated regulation of and binding to PU.1 and MITF transcription factors, but the precise fusion mechanism remains unclear. Understanding the underlying mechanisms of cell fusion is important for the development of therapeutic strategies to treat osteoclast-related bone diseases.

However, it has been ascertained that macrophages are heterogeneous based on their origin and niche [17,18], and not all cells are synchronized; specifically, some cells fuse and develop into multinucleated giant cells, but many remain as monocytes, during osteoclast differentiation in vitro. Thus, in this study, to determine the transcriptomic differences between these unsynchronized cells, which can be morphologically classified as single nucleated cells (SNCs) and MNCs upon RANKL administration, and to perform a comparison with bone marrow-derived macrophages (BMMs), we investigated differentially expressed genes (DEGs) between SNCs and MNCs using RNA sequencing (RNA-seq). Principal component analysis (PCA) clearly separated the overall transcriptome profiles of SNCs and MNCs. By comparing the DEGs in each group, we found increased expression of seven genes (*Aif1*, *Calcrl*, *Gsta3*, *Ifit2*, *Marco*, *Tmem26*, and *Ube3a*) in MNCs, and the expression of these genes was decreased in SNCs. Our findings demonstrate that MNC-specific highly expressed genes and heterogeneous cell populations induce osteoclast fusion and maturation.

## 2. Results

### 2.1. SNCs and MNCs Have Distinct Transcriptomic Profiles during Osteoclastogenesis

To determine the differences between SNCs and MNCs, we performed RNA-seq analysis. We isolated murine BMMs and cultured them for 4 days with RANKL and M-CSF (Figure 1A). SNCs were collected via rough dissociation, and the attached MNCs were washed thoroughly to prevent contamination. The MNCs were directly lysed for RNA extraction. RNA-seq analysis was performed on BMMs, SNCs, and MNCs to determine the DEGs. The number of reads generated ranged from 73,952,744 to 87,103,520 genes, and the trimmed clean reads were mapped to the mouse reference genome with high alignment rates (Table 1). The results of PCA showed that the overall transcriptome profiles of BMMs and SNCs were clearly separated from those of MNCs (Figure 1B), at least implicitly suggesting that SNCs may comprise the middle stage of the osteoclast lineage or represent a slightly different committed fate, as compared to MNCs in the late stages of osteoclast differentiation.

The DEGs were analyzed in several ways based on comparisons between two groups as follows: BMMs and SNCs, BMMs and MNCs, and SNCs and MNCs. We generated a Venn diagram and scatter plots to depict the overlaps and DEGs in these sets (Figure 1C,D). As expected, the number of DEGs between SNCs and MNCs was fewer than that between the other two comparisons, suggesting that SNCs are closer to MNCs than to BMMs.

By overlapping the entire set of DEGs from all three datasets, a total of 1228 common DEGs were identified (Figure 1C). We hypothesized that these genes are potential targets for osteoclast differentiation. Of these 1228 DEGs, 719 genes were upregulated, and 509 genes were downregulated in MNCs compared to their expression in SNCs. When the SNC–MNC set was compared with the other two comparison sets, 97% of the 509 MNC-specific downregulated genes were upregulated in both SNCs and MNCs, compared to their expression in BMMs (Figure 2A). Furthermore, 719 MNC-specific upregulated genes were downregulated in SNCs and MNCs compared to those in BMMs, comprising 93.2% and 92.2% of DEGs, respectively. Most of the upregulated genes in MNCs, compared to their expression in SNCs, revealed patterns of downregulation in the BMM–SNC and BMM–MNC comparisons, suggesting that these genes were extremely up- or down-regulated in SNCs versus MNCs. Thus, SNCs and MNCs are two distinct subsets that differ in their characteristics. Using GO analysis, we found that the 719 MNC-specific upregulated genes were involved in cell cycle and inflammatory response biological processes, whereas the 509 MNC-specific downregulated genes were involved in oxidation-reduction and metabolic processes (Figure 2B). Together, these results suggest that SNCs are more active and motile than MNCs, which attach to and stack on the bone for resorption.

### 2.2. SNCs Include Preosteoclast Cells Committed to Osteoclasts

To determine whether SNCs are preosteoclasts that are more differentiated than BMMs but less differentiated than MNCs, we assessed the osteoclast differentiation status of SNCs. We confirmed the expression levels of 43 canonical osteoclast-related genes in RNA-seq data, which were significantly differentially expressed in at least one comparison set (Figure 3A). Of interest, many genes (25 of 43), including ATPase H+ transporting accessary protein 1 (*Atp6ap1*), *DC-STAMP*, *OC-STAMP*, *NFATc1*, *NF-kB,* and TNF receptor-associated factor 6 (*TRAF6*), were highly expressed in SNCs rather than in MNCs. We further validated their expression levels using qRT-PCR (Figure 3B). Some genes, such as *NFATc1*, *DC-STAMP,* and calcitonin receptor (*Calcr*), showed higher expression in SNCs than in MNCs, consistent with the RNA-seq results. Otherwise, integrin beta-3 (*Itgb3*), *c-Src*, *OC-STAMP*, and *Oscar* were maintained at higher levels in SNCs and MNCs.

To determine whether SNCs represented preosteoclasts immediately before fusion, non-adherent cellss were collected 2 days after RANKL stimulation and then recultured in RANKL-containing medium (Figure 4A). Although the cells were not adherent in the previous culture conditions, they attached in the new environment. After 2–3 additional days, the cells differentiated into fused TRAP-positive cells, suggesting that SNCs can differentiate into MNCs. To test our hypothesis, we analyzed cell fusion by separating adherent and non-adherent cells. The two types of cells were separated after 2 days of RANKL stimulation followed by staining with two distinct cell trackers, CMFDA or DiI, and then co-cultured to investigate cell fusion (Figure 4B). When the two adherent cell types were mixed, the cells fused and developed into multinucleated giant cells within 2–3 days of RANKL stimulation. However, two non-adherent cell types did not fuse well. Heterogeneity between two cells is critical for fusion during osteoclast differentiation [19]. To investigate whether the adherent and non-adherent cells stimulate each other, the DiI-stained adherent or suspended cells were co-cultured with non-stained adherent cells (Figure 4C). Here, more fused cells were detected in the adherent cell–non-adherent cell co-culture than in the adherent cell culture, and these fused within 1–2 days. These results suggest that non-adherent SNCs more actively form contacts with adherent MNCs to become fused giant cells. Additionally, cell fusion occurred slowly between homogeneous cell co-cultures with both MNCs or both non-adherent SNCs. Altogether, these results suggest that SNCs may be osteoclast-committed cells that express canonical osteoclast genes.

During osteoclast differentiation, the expression of proteins participating in the cell cycle is reduced to save energy [20]. The cell cycle is the most significantly regulated process in both SNCs and MNCs. In general, cells cease proliferation to initiate differentiation [21]. Therefore, we examined cell proliferation in SNCs and MNCs by detecting Ki67 expression (Figure 4D). As assumed, Ki67-positive cells were all SNCs, but none of the MNCs were proliferating cells. Therefore, SNCs actively proliferated, although the expression of cell cycle-related genes was decreased in SNCs, and these genes were required for osteoclast fusion.

### 2.3. MNC-Specific DEGs Are Involved in Disease-Related Pathways

To dissect the DEGs between SNCs and MNCs, we focused on 2657 DEGs from the SNC-MNC comparison set (Figure 1C). Among the DEGs, 68% (1805 genes) were upregulated in MNCs, whereas 32% (852 genes) were upregulated in SNCs. In the GO analysis, 1805 genes were mostly associated with cell cycle- and division-related biological processes (Figure 5A). However, 852 genes were mainly related to the mitochondria, metabolic processes, and membrane cellular components. To determine whether SNCs comprise a lineage distinct from BMMs, which is also different relative to the relationship between MNCs and BMMs, we further performed a comparison of 2657 DEGs based on the BMM-SNC and BMM-MNC comparison sets. Here, 881 genes were specific DEGs in the BMM-SNC comparison set, and these were not different in the BMM-MNC comparison set. In contrast, there were only 203 DEGs in the BMM-MNC comparison set that were not DEGs in the BMM-SNC comparison set (Figure 1C). To determine the importance of these DEGs, we performed GO and KEGG analyses with the 881 and 203 DEGs. As shown in Figure 5B, 881 DEGs were related to cell cycle and nucleocytoplasmic transport. Of the 203 genes, eight (including *Ccl5*, *Cxcl1*, *Fos*, and *Cxcl2*) belonged to the rheumatoid arthritis and TNF signaling pathways, which are associated with osteoclasts. These findings indicate that MNC-specific DEGs are associated with disease and immune response pathways, whereas SNC-specific DEGs are involved in cellular mechanisms.

Additionally, 345 DEGs in the SNC-MNC comparison set were not significantly altered in the BMM-SNC and BMM-MNC comparison sets that were enriched in the biological process of the spliceosome (Figure 5B).

### 2.4. Calcrl, Marco, Tmem26, and Ube3a Control SNCs and MNCs

Next, we sought to determine the gene that regulates osteoclast fusion, resulting in mature osteoclasts, defined as MNCs. Therefore, we analyzed the expression patterns between BMMs and SNC and between BMMs and MNCs. Only seven genes revealed opposite expression patterns for which levels were significantly decreased in SNCs compared to those in BMMs, whereas these levels were significantly increased in MNCs compared to those in BMMs. The remaining 3476 genes showed the same expression patterns, both decreased (1786 genes) or increased (1690 genes), in SNCs and MNCs compared to those in BMMs. We further validated the seven genes using qRT-PCR. All genes except *Gsta3* and *Ube3a* were significantly upregulated in MNCs in accordance with RNA-seq results, but their expression levels were diminished in SNCs (Figure 6A,B, Table 2).

Allograft inflammatory factor 1 (Aif1) is a pan marker of macrophages [22] that is upregulated by RANKL stimulation during osteoclast differentiation [23]. Calcitonin receptor-like (Calcrl) protein is a G-protein-coupled neuropeptide receptor that regulates blood pressure, angiogenesis, cell proliferation, and apoptosis [24,25]. Calcrl is expressed in hematopoietic cells and has a role in malignant hematopoietic cells [25,26]. Transmembrane 26 (Tmem26) is a surface marker protein of beige adipocytes, and M2 macrophage activation contributes to beige fat development [27,28]. Glutathione S-transferase alpha 3 (Gsta3) is a member of the GST family and is a detoxification enzyme that protects against oxidative stress [29]. Increased oxidative stress promotes osteoclastic bone remodeling and decreased bone loss [30]. Interferon-induced protein with tetratricopeptide repeats 2 (Ifit2) is involved in the innate immunity-mediated regulation of genes in response to interferon stimulation and is important for the fracture healing process in osteoporosis [31]. Macrophage receptor with collagenous structure (Marco) is a member of the class-A scavenger receptor family involved in innate immunity to bacterial infection [32]. Marco is expressed on macrophages, dendritic cells, and inflammatory monocytes [33]. Ubiquitin protein ligase E3a (Ube3a) is an E3 ubiquitin ligase that attaches ubiquitin to its specific substrate proteins for degradation via the ubiquitin-proteasome system [34]. Ube3a is important in neurodevelopment and cancer, but it is not well-studied in osteoclastogenesis.

To examine whether these seven genes regulate osteoclastogenesis, especially cell fusion, we first assessed their role using a CRISPR-Cas9 genome-editing tool to generate RAW264.7 cells lacking each gene. We selected cells in which each gene was deleted completely by confirmation using deep-sequencing analysis (Table 3). *Aif1*, *Gsta3*, and *Ifit2* gene-knockout (KO) cell lines could not be obtained, even though we combined two gRNAs.

The KO cells were cultured with RANKL to assess fusion. The cells lacking *Calcrl*, *Marco*, and *Ube3a* did not differentiate into MNCs and showed a reduced number of TRAP-positive cells (Figure 6C,D). Although the deficient cells were not fused, some revealed TRAP-positive signaling in SNCs. Therefore, we determined bone resorption activity in the KO cell lines. *Calcrl*- or *Ube3a*-deficient cells had markedly reduced resorption activity (Figure 6C,D). However, *Marco*-deficient cells showed slight resorption activity. *Tmem26*-deficient cells further differentiated compared to the controls and presented with highly enhanced bone resorption activity, suggesting that *Tmem26* functions as a negative regulator in MNCs to prevent SNC-specific signaling.

### 2.5. Calcrl Is Critical for MNC Development

*Calcrl*-KO inhibited osteoclast fusion completely, and *Calcrl* was one of the most highly expressed genes in MNCs compared to its levels in SNCs (Figure 6). Therefore, we examined Calcrl expression during osteoclast differentiation (Figure 7). Calcrl protein was highly expressed in MNCs (Figure 7A). To determine Calcrl expression levels during osteoclast fusion, Western blotting was performed on a *Calcrl*-KO cell line that was not able to develop fused osteoclasts. RAW 264.7 cells fused and developed MNCs within 3–4 days of RANKL stimulation. As shown in Figure 7B, Calcrl protein was detected in control cells on days 3 and 4, whereas it was not detected on day 1 and 2. We then tested mRNA expression levels of osteoclast-related genes in *Calcrl*-KO cells (Figure 7D). The expression levels of osteoclast-specific genes, such as *Oscar* and *Acp5*, were significantly decreased in *Calcrl*-KO cells compared to those in controls on day 3. mRNA levels of transcription factors, such as *c-Fos* and *Jun*, were altered in *Calcrl*-KO cells. Taken together, these results demonstrate that Calcrl is necessary for cell–cell fusion and may regulate transcription factors during osteoclastogenesis.

## 3. Discussion

Bone marrow-derived monocytes are heterogeneous and contain many hematopoietic precursors, leading to the generation of diverse cells, including dendritic cells, myeloid blasts, and macrophages [35]. Osteoclasts differentiate from BMMs with M-CSF and RANKL stimulation. Although RANK-positive cells can differentiate into mature osteoclasts in vitro, which are TRAP- and CtsK-positive, and bone-resorbing cells, they have different phenotypes and functions. When the cells are faced with the same stimulation conditions for osteoclasts, not all cells develop into multinucleated giant osteoclasts, with many remaining as mononuclear cells. Therefore, it is unclear how they choose their fate or what is the deciding factor for cell–cell fusion and selecting their fusion partners. In this study, we identified the factors regulating osteoclast fusion via RNA-seq analysis in SNCs and MNCs with RANKL stimulation. We selected DEGs between SNCs and MNCs and then examined their function in KO cell lines. *Calcrl*-, *Marco*-, or *Ube3a*-deficient RAW 264.7 cells did not differentiate into mature osteoclasts upon RANKL stimulation, suggesting that these are novel genes controlling osteoclast differentiation.

To determine the genes involved in cell fusion, we analyzed the DEGs in several ways. The DEGs were compared with reported macrophage- and osteoclast-specific genes (KEGG database, http://www.genome.jp (accessed on 8 July 2021), [36]). Levels of osteoclast-specific genes were increased in SNCs compared to those in MNCs (Figure 3). It is suggested that SNCs gain osteoclastogenic activity even if the cells are not yet fused. This is related to previous studies suggesting that non-fused osteoclasts can resorb bone at low levels [15]. Since we focused on the cell fusion mechanism, DEGs with increased expression in MNCs compared to levels in SNCs were selected. The seven DEGs presenting opposite expression patterns in SNCs and MNCs were our first targets. The expression of Marco was most highly decreased in SNCs compared to that in MNCs. However, *Marco*-deficient RAW 264.7 cells were able to fuse, although the fusion rate was significantly lower than that of controls. It is possible that Marco is not a direct regulator or is not sufficient in controlling cell fusion. Calcrl and its co-receptors, namely receptor activity modifying protein (RAMP) 1, RAMP2, or RAMP3, bind calcitonin gene-related peptide (CGRP) or adrenomedullin (ADM) [37]. Calcrl is expressed in the membranes of osteoclast precursors, but the ADM- and CGRP-dependent function of Calcrl during osteoclastogenesis remains ambiguous [38]. Here, we clarified a novel function of Calcrl as a key factor for osteoclast fusion, as our findings suggested that cell fusion was blocked in *Calcrl*-deficient RAW 264.7 cells. We did not examine the function of CGRP or ADM in the *Calcrl*-deficient cells, and the Calcrl-related cell function could not be assessed based on the typical receptor and ligand mechanism. Ube3a is homologous to the E6AP C-terminus (HECT) E3 ligase and is a critical factor for normal neurodevelopment, as its loss of function in the brain leads to Angelman syndrome [39]. However, its function in the bone has not yet been studied. Ube3a was not highly upregulated or downregulated during osteoclastogenesis (Table 2, Figure 6B), although *Ube3a* deficiency in RAW 264.7 cells ablated cell-cell fusion. Our results suggest a novel strategy for osteoclast differentiation mediated by Ube3a.

We used a gRNA library and designed new gRNAs to delete the target genes in a RAW 264.7 mouse macrophage cell line using a CRISPR-Cas9 genome-editing tool. However, *Aif1*-, *Gsta3*-, or *Ifit2*-KO cell lines could not be developed. We will thus need to test other gRNAs to delete these genes; alternatively, it is possible that these genes are involved in cell survival in RAW 264.7 cells.

BMMs proliferate in the presence of M-CSF, whereas fused osteoclasts are post-mitotic with RANKL stimulation [40]. In general, cell differentiation is tightly coupled with a permanent exit from the cell cycle [41]. During osteoclast differentiation, proliferating cells are detected at the early phase, followed by G1 arrest during the development of mature osteoclasts with multiple nuclei. Rahman et al. demonstrated that with the inhibition of DNA synthesis in the first proliferative phase, osteoclastogenesis is completely suppressed [40]. Therefore, DNA synthesis is required in the early phase. We suggested that downregulated cell cycle-related gene expression in SNCs leads to these cells remaining as SNCs without fusion. However, if the SNCs are replaced in the new environment, they can develop into multinuclear osteoclasts. Mature osteoclasts exhibit reduced expression of cell cycle- and DNA repair and replication-related genes to save energy consumption, as compared with that in immature and precursor cells [20]. Cell death signaling mediated by apoptosis is induced in mature osteoclasts followed by the downregulation of ATP [20]. An et al. showed that metabolism-related processes, such as the TCA cycle, are increased in multinuclear osteoclasts based on a comparison among precursor cells, intermediate osteoclasts, and multinuclear osteoclasts derived from RAW 264.7 cells. [42]. However, the levels of metabolic process- and transport-related genes were downregulated in MNCs compared to those in SNCs. It is possible that our SNCs are closer to multinuclear osteoclasts differentiated from RAW 264.7 cells, and accordingly, proteomics is suggested as a future study direction to address this.

Macrophages are essential for the bone remodeling process [43]. They are monocytic cells involved in inflammation that participate in limb regeneration and the production of osteoblasts [44,45,46]. Macrophage polarization results in a function associated with pro- or anti-inflammatory effects, and these cells are classified as classically activated (M1) or alternatively activated (M2) [47]. M2 macrophages are induced by IL-4, IL-10, or RANKL, whereas M1 macrophages are induced by IFN-γ or LPS. These two macrophage types have specific phenotypes [48,49]. However, macrophage polarization is not well-defined because M1 and M2 macrophages switch in response to the local microenvironment [50], and M1 macrophages are also involved in bone formation and osteoclastogenesis [18]. Jeganathan et al. [51] suggested that MNCs are heterogeneous and can be committed to the osteoclast lineages but can also develop into other types of MNCs based on gene expression and cytoskeletal rearrangements. It is possible that SNCs and MNCs are differently committed cells and represent a source of heterogeneity in several lineages.

Several molecular mediators have been reported to be important for osteoclast fusion, including CD47, syncytin 1, DC-STAMP, and CD44 [52,53,54]. Although DC-STAMP is a well-known transmembrane protein involved in osteoclast fusion, its ligand has not yet been defined. To determine the expression levels of *Calcrl*, on day 4, we isolated mRNAs from SNCs or MNCs. The mRNA expression levels were significantly higher in MNCs than SNCs (Figure 6B). Our protein expression data showed high expression only on day 4, as transcripts expression, even though it was analyzed in the whole-cell extracts, was not separate between SNCs and MNCs (Figure 7B). These results suggest that the translation of Calcrl protein begins with cell fusion. Cell fusion is a phenotype of mature osteoclasts; however, fusion is not an indicator of osteoclast function. TRAP-positive cells were detected in the SNCs based on our data. *DC-STAMP*- or *OC-STAMP*-deficient mice present have low levels of bone-resorbing activity [55,56]. Thus, fusion-regulating genes are not identical to osteoclast marker genes. McDonald et al. revealed that mature osteoclasts are recycled via osteomorphs and undergo fission to form small, motile cells from the multinucleated osteoclasts [3]. We speculate that the SNCs in our study have some resemblance to osteomorphs.

Recent studies have demonstrated that heterogeneity is required for osteoclast fusion. The DC-STAMP expression level should be different between the two fusion partners [57]. CD47 and syncytin-1 are highly expressed in monocytes compared to their levels in multinucleated macrophages [57]. Cell fusion occurs between two heterogeneous cells. We tested cell fusion between SNCs and MNCs (Figure 4). Mature osteoclasts were well-developed in cultures comprising SNCs and MNCs, as compared to those in homogeneous cultures of SNCs or MNCs alone. These results suggest that heterogeneity is necessary for osteoclast fusion and maturation via RANKL stimulation. The specific regulation of new markers and heterogeneity of the cell population during osteoclastogenesis could provide an alternative regulatory mechanism in osteo-related diseases. Additionally, these results shed light on surface makers that could be used to distinguish preosteoclasts from mature osteoclasts.

## 4. Materials and Methods

### 4.1. Cell Culture and Osteoclast Differentiation

BMMs were isolated from the long bones (femur and tibia) of female mice, as described previously [58]. For osteoclast differentiation analysis, BMMs were incubated in osteoclastogenic medium containing 30 ng/mL M-CSF and 50 ng/mL RANKL (Peprotech, Rocky Hill, NJ, USA) for 4–6 days. After osteoclast differentiation, SNCs were collected using cell dissociation buffer (Millipore Sigma, Burlington, MA, USA) treatment, and MNCs were directly lysed after thorough washing with PBS.

Murine macrophages (RAW 264.7) were cultured in αMEM medium containing 10% FBS (Gendepot, Katy, TX, USA). RAW264.7 cells were cultured for 4–6 days with 50 ng/mL RANKL for differentiation into osteoclasts. The differentiated cells were fixed with 4% formaldehyde (Duksan Company, Gyeonggi, Korea) for 10 min and stained with a TRAP staining kit (Cosmo Bio, Tokyo, Japan) according to the manufacturer’s instructions. For the visualization and quantification of TRAP staining, whole plates were scanned and analyzed under a Lionheart FX (BioTek, Winooski, VT, USA) fluorescence microscope with the Gen 5 software. This study was carried out in strict accordance with the recommendation of the Guide for the Care and Use of Laboratory Animals of the Chonnam National University (CNU IACUC-YB-2019-49).

### 4.2. Bone Resorption Assay

Bone resorption assays were performed using a Bone resorption assay kit (Cosmo Bio, Tokyo, Japan), as described previously [59]. Briefly, RAW 264.7 cells and knockout cells were cultured for 7 days in 100 ng/mL RANKL-containing medium. The pit area was analyzed using the ImageJ software.

### 4.3. RNA Isolation and Quantitative Real-Time PCR (qRT-PCR)

RNA was isolated from BMMs, SNCs, and MNCs, and qRT-PCR was performed as described previously [11]. Briefly, RNA was isolated using an RNeasy kit (Qiagen, Hilden, Germany), and cDNA was synthesized using a Prime Script RT reagent kit (Takara, Kusatsu, Shiga, Japan) following the manufacturer’s instructions. qRT-PCR was performed using the Quant Studio 3 real-time PCR system with Power SYBR Green PCR Master Mix (Applied Biosystems, Waltham, MA, USA) and the primers listed in Table 4. All experiments were performed in duplicate, and the results were normalized to the levels of the 18S gene using the 2^−∆∆Ct^ method for data analysis.

### 4.4. Immunohistochemistry and Western Blotting

BMMs and osteoclasts were cultured directly under osteoclastogenic conditions for the indicated times. To assess fusion, a fusion assay was performed via cell labeling [60]. Briefly, the BMMs were cultured for 2 days under osteoclastogenic conditions. Cells in suspension subjected to RANKL stimulation were collected using cytocentrifugation. Adherent cells were directly labeled on the plate for 30 min and were then scraped off. Cells were labeled with Cell Tracker Green CMFDA or Cell Tracker CM-DiI (Thermo Scientific, Waltham, MA, USA). Differently labeled cells were mixed and co-cultured, or the DiI-labeled cells were co-cultured with non-labeled adherent cells for 2–3 days in an osteogenic medium. To determine cell proliferation, a Ki67 antibody (Novus, Littleton, CO, USA) was used. Cells were counterstained with DAPI, purchased from Thermo Fisher Scientific (Waltham, MA, USA), according to the manufacturer’s instructions. Osteoclasts were imaged using a Carl Zeiss microscope (Carl Zeiss, Jena, Germany). Protein isolation and Western blotting were performed as previously described [11]. A polyclonal Calcrl antibody (Sigma-Aldrich, St. Louis, MO, USA) was used for immunostaining and Western blotting.

### 4.5. RNA Sequencing and Data Analyses

BMMs were cultured for 4–5 days in M-CSF- and RANKL-containing medium to differentiate them into osteoclasts. When fused osteoclasts were observed, the SNCs and MNCs were harvested simultaneously. First, the SNCs were detached with a cell dissociation solution (Millipore Sigma, Burlington, MA, USA), and the remaining cells were washed with PBS several times to remove the SNCs. Since the MNCs were strongly attached to the cell culture plates, they were directly lysed for RNA extraction. RNA was isolated using the RNeasy Mini kit (Qiagen, Hilden, Germany), and quality control and sequencing were performed by Macrogen Incorporated (Seoul, Korea). Briefly, cDNA was transcribed using SuperScript II reverse transcriptase (Thermo Fisher Scientific, Waltham, MA, USA), and a cDNA library was prepared using the TruSeq Stranded mRNA LT Sample prep kit (Illumina Inc., San Diego, CA, USA).

All raw sequence reads were preprocessed using Trimmomatic (version 0.39) [61] to remove adapter sequences and bases with low sequencing quality. The remaining clean reads were mapped to the mouse reference genome (mm10) using HISAT2 (v2.1.0) [62] with the default parameters. BAM files generated from HISAT2 were further processed with Cufflinks (v2.2.1) [63] to quantify transcript abundances using the fragment per kilobase of exon per million fragments mapped (FPKM) normalization method. Differential expression analysis, performed using Cuffdiff (v2.2.1), was subsequently employed to analyze DEGs with an FPKM value > 1 in at least one sample and a q-value < 0.05. We performed Gene Ontology (GO) category and Kyoto Encyclopedia of Genes and Genomes (KEGG) pathway enrichment analyses of the DEGs using the DAVID functional annotation tool (https://www.david.ncifcrf.gov, accessed on 8 July 2021). The mouse reference genome sequence and annotation data were downloaded from the UCSC genome browser (https://www.genome.ucsc.edu, accessed on 8 July 2021), and the R software was used to visualize the results.

### 4.6. CRISPR-Cas9 Genome-Editing

To delete the target genes, guide RNAs (gRNAs) for the target genes were designed using Cas-Designer (https://www.rgenome.net, accessed on 8 July 2021). The gRNAs were synthesized via in vitro transcription using T7 RNA polymerase (New England Biolabs, Ipswich, MA, USA). RAW 264.7 cells were transfected with the CRISPR/Cas9 RNP complex and gRNAs using the Neon Transfection System and transfection kit (Thermo Fisher Scientific) following the manufacturer’s instructions. The transfected cells were selected as single colonies, and gene deletion was confirmed using deep sequencing. The genomic DNA was extracted and then amplified using Ex Taq Polymerase (Takara, Kusatsu, Shiga, Japan). The PCR fragments were amplified using HT Dual index-containing primers and sequenced on an Illumina MiniSeq platform. The sequencing data were analyzed using the RGEN tool [64].

### 4.7. Statistical Analyses

All data are presented as the mean ± standard deviation of three independent experiments or as otherwise indicated. Significance was analyzed using unpaired two-tailed Student’s *t* tests and highlighted based on the following notations: * *p* < 0.05; ** *p* < 0.01.

## Figures and Tables

**Figure 1 ijms-23-06421-f001:**
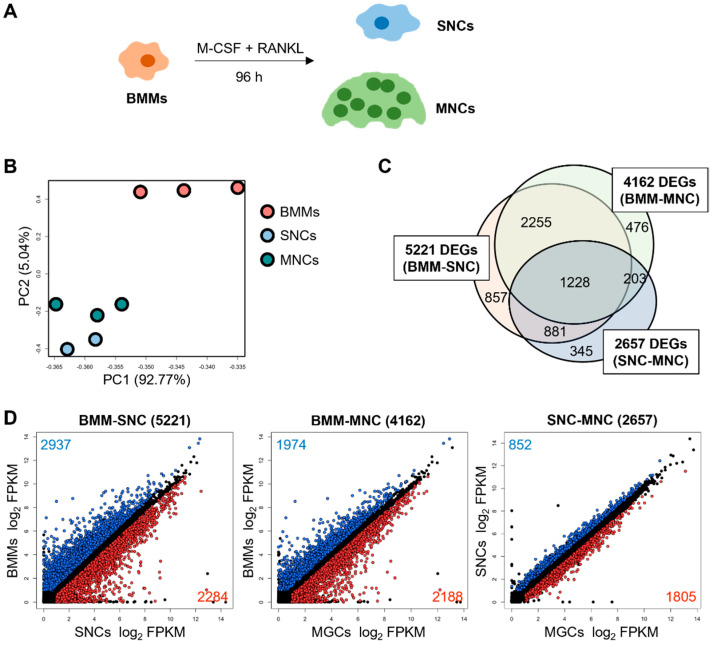
Transcriptomic profiling of single nucleated cells (SNCs) and multinucleated cells (MNCs) reveals distinct populations during osteoclastogenesis. (**A**) Schematic diagram of osteoclast differentiation and isolation of bone marrow-derived macrophages (BMMs), SNCs, and MNCs. (**B**) Principal component analysis (PCA) of BMMs, SNCs, and MNCs. Each dot represents the expression profile of one sample: *n* = 2–3. (**C**) The Venn diagram indicates differentially expressed genes (DEGs) for the three comparisons, specifically BMM-SNC, BMM-MNC, and SNC-MNC. (**D**) The DEGs for the three comparisons are displayed on the scatter plot. Blue and red indicate significantly dysregulated genes, and black indicates no significant differences (fragments per kilobase of exon per million fragments mapped (FPKM) > 1, q-value < 0.05).

**Figure 2 ijms-23-06421-f002:**
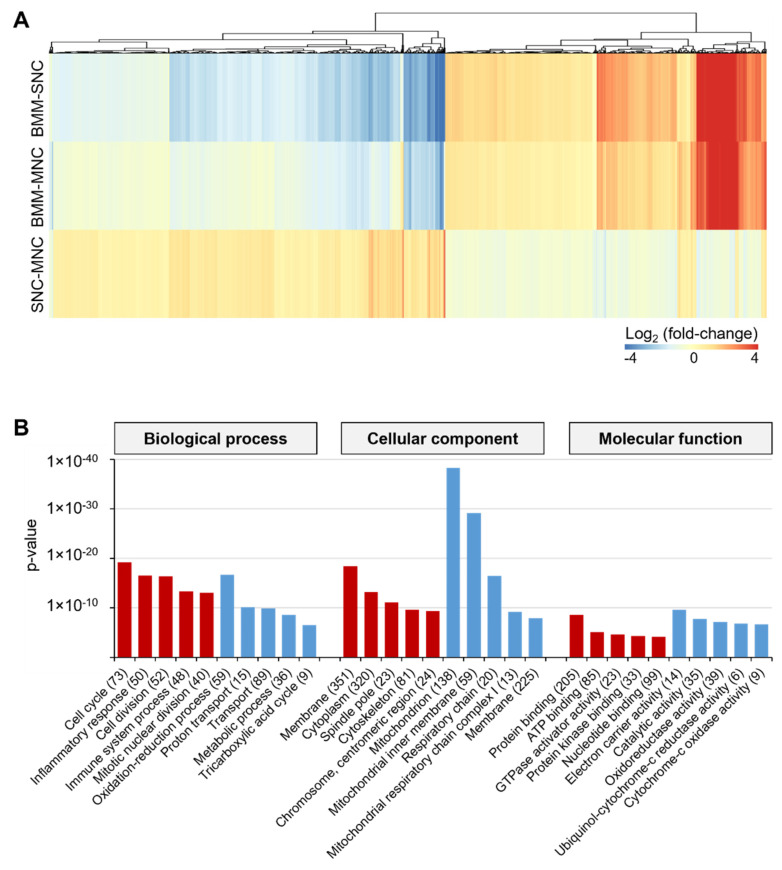
Depiction of 1228 DEGs common to all comparisons. (**A**) Heatmap analysis indicating the highly upregulated or downregulated genes from 1228 common genes in the SNC-MNC comparison (q-value < 0.05). (**B**) Gene ontology (GO) analysis results for the 1228 overlapping genes. Blue bars indicate SNC-specific upregulated genes, whereas red bars indicate MNC-specific upregulated genes from the comparison between SNCs and MNCs.

**Figure 3 ijms-23-06421-f003:**
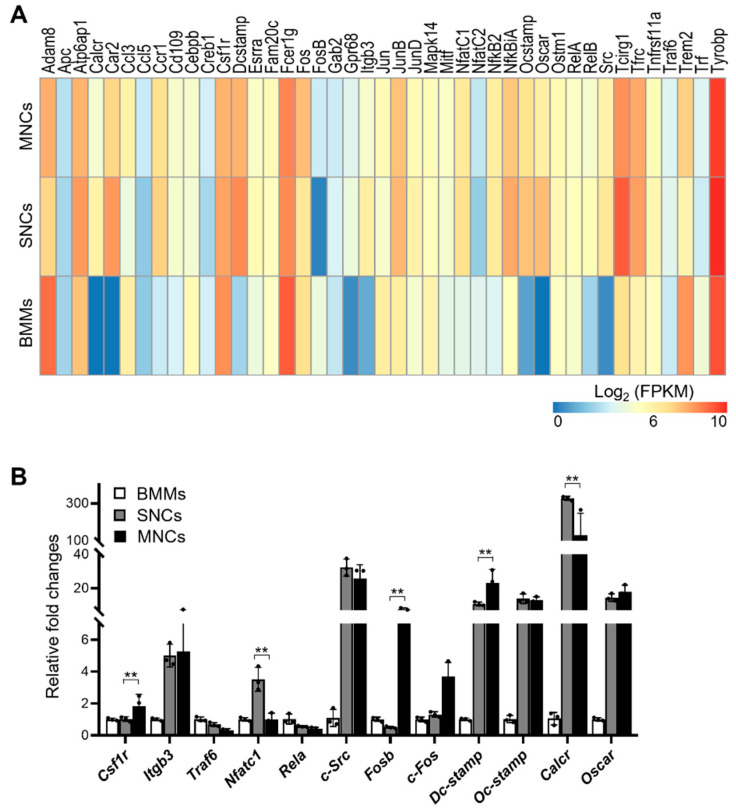
Osteoclast-related genes are highly expressed in SNCs. (**A**) Canonical osteoclast differentiation-associated genes are revealed by a heatmap based on the RNA-seq analysis. The color bar units represent the log_2_ units in FPKM; q-value < 0.05. (**B**) qRT-PCR was performed on BMMs, SNCs, and MNCs to validate RNA-seq analysis (**, *p* < 0.01).

**Figure 4 ijms-23-06421-f004:**
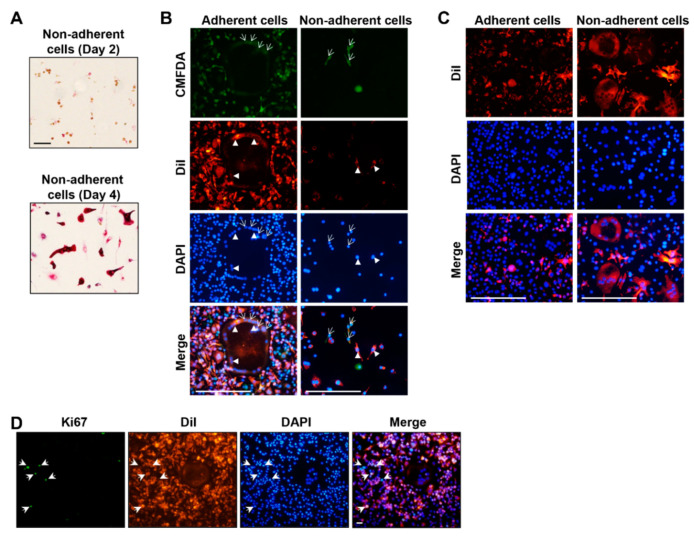
SNCs are osteoclast-committed cells. (**A**) Suspended SNCs were collected on day 2 following RANKL stimulation and then recultured in RANKL-containing medium for an additional 2 days (day 4). The cells on days 2 and 4 were fixed and stained with TRAP to determine the mature osteoclasts. Scale bar, 100 µm. (**B**) Adherent cells or non-adherent cells were collected on day 2 of RANKL stimulation. Two cell populations stained separately with CMFDA (arrows) or DiI (arrow heads) were co-cultured for 3 days in RANKL-containing medium. The cells were fixed, and fusion was detected. Scale bar, 100 µm. (**C**) DiI-stained adherent (left panel) or non-adherent (right panel) cells on day 2 following RANKL stimulation were co-cultured with unstained adherent cells for 2 days with RANKL stimulation Scale bar, 100 µm. (**D**) Cell proliferation was determined based on Ki67 (arrows) immunostaining on day 4 after RANKL stimulation. Scale bar, 20 µm.

**Figure 5 ijms-23-06421-f005:**
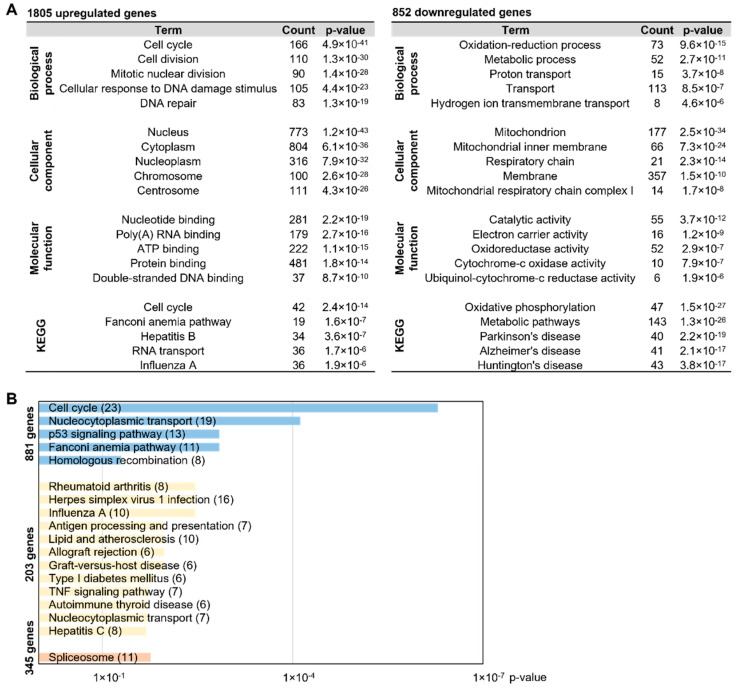
MNC-specific DEGs are involved in disease-related pathways. (**A**) Of the 2657 DEGs from the SNC-MNC comparison, 1805 upregulated and 852 downregulated genes in MNCs were analyzed via GO analysis. The top five highly significant GO terms are revealed. (**B**) Among the DEGs in Figure 1C, 881, 203, and 435 genes were analyzed by GO analysis (q-value < 0.05).

**Figure 6 ijms-23-06421-f006:**
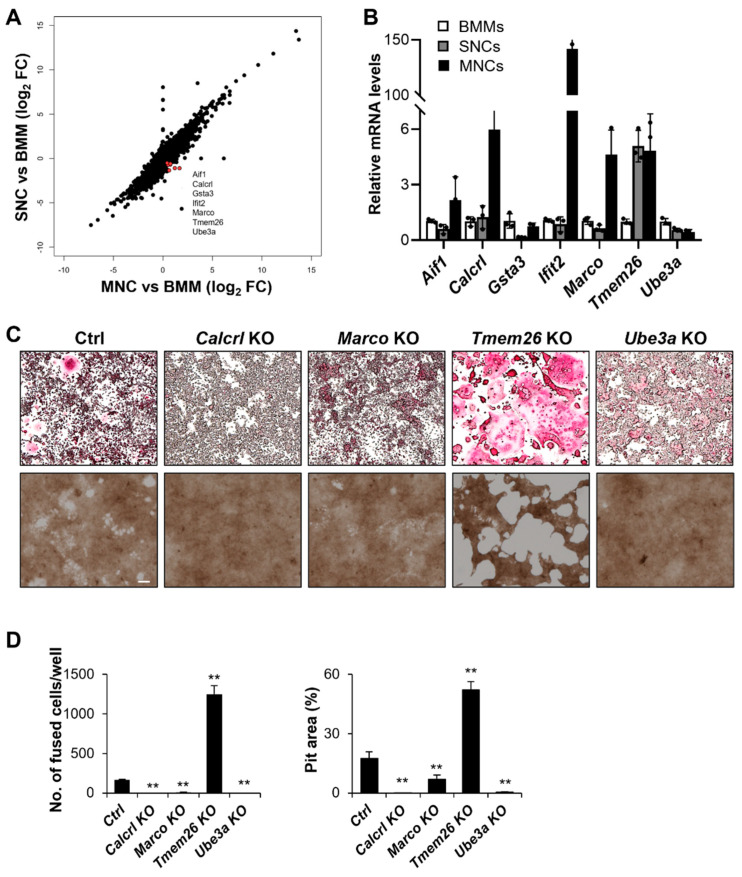
Seven potential targets regulate SNCs and MNCs. (**A**) In the SNC-BMM and MNC-BMM comparisons, only seven genes showed an opposite direction (red dots). Other genes revealed the same expression patterns (black dots). (**B**) The mRNA expression levels of the seven genes were tested using qRT-PCR. (**C**) TRAP staining (up) and bone resorption (down) assays were performed to examine *Calcrl-*, *Marco-*, *Tmem26-*, or *Ube3a*-knockout (KO) RAW 264.7 cells. Cells were cultured in RANKL-containing medium to differentiate them into osteoclasts. Ctrl, control RAW 264.7 cells. Scale bar, 100 µm. (**D**) The fused osteoclasts with more than three nuclei in (**C**) were counted after TRAP staining (**right** graph). The fit area in (**C**) was measured using the ImageJ software (**left**) (**, *p* < 0.01 compared to Ctrl).

**Figure 7 ijms-23-06421-f007:**
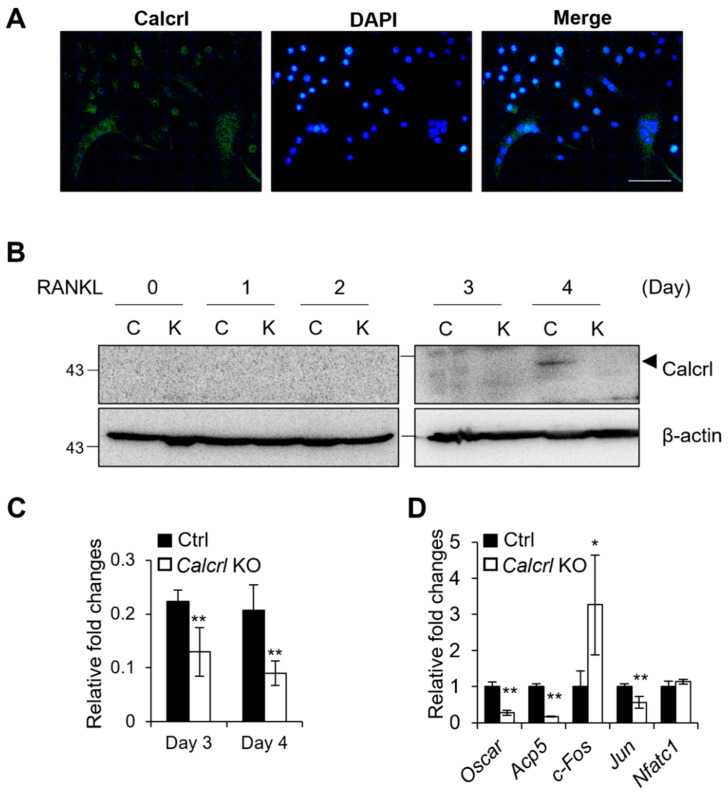
Calcrl is critical for MNC development. (**A**) Calcrl expression in the osteoclasts was detected with an immunofluorescence assay. BMMs were cultured for 4 days with M-CSF and RANKL stimulation. Scale bar, 100 µm. (**B**) Calcrl protein expression was detected using Western blotting on Ctrl (C) and *Calcrl*-knockout (KO or K) RAW 264.7 cells. The cells were differentiated into osteoclasts via RANKL stimulation. (**C**) The graph shows quantified Calcrl protein levels on Day 3 and 4 using Image J software (*n* = 3). (**D**) The mRNA expression of osteoclast-related genes was examined using qRT-PCR in Ctrl and *Calcrl*-KO osteoclasts. The cells were harvested on day 3 after RANKL stimulation (*, *p* < 0.05; **, *p* < 0.01 compared to Ctrl).

**Table 1 ijms-23-06421-t001:** RNA-seq analysis statistics.

Sample	Number of Reads (Sum of Pairs)	Number of Reads after Trimming	Alignment Rate (%)
BMM 1	79,081,594	77,290,554	98.62%
BMM 2	73,952,744	72,383,444	98.17%
BMM 3	84,004,640	81,523,436	98.15%
MNC 1	75,068,812	73,463,268	98.82%
MNC 2	79,729,144	77,767,782	98.55%
MNC 3	81,017,248	79,417,460	98.74%
SNC 1	84,672,352	82,673,110	98.45%
SNC 2	87,103,520	85,087,954	98.45%

**Table 2 ijms-23-06421-t002:** Seven DEGs revealed with opposite expression patterns between the BMM-SNC and BMM-MNC comparisons.

Gene	BMM–SNC		BMM–MNC		Description
	Log_2_ FC	q-value	Log_2_ FC	q-value	
*Aif1*	−0.75	0.0096	0.77	0.0022	Allograft inflammatory factor 1
*Calcrl*	−1.41	0.0003	0.64	0.0020	Calcitonin receptor-like
*Gsta3*	−1.33	0.0003	1.30	0.0003	Glutathione S-transferase, alpha 3
*Ifit2*	−0.90	0.0009	0.59	0.0335	Interferon-induced protein with tetratricopeptide repeats 2
*Marco*	−1.82	0.0003	1.98	0.0003	Macrophage receptor with collagenous structure
*Tmem26*	−0.80	0.0072	0.91	0.0003	Transmembrane protein 26
*Ube3a*	−0.55	0.0138	0.44	0.0297	Ubiquitin protein ligase E3A

FC, fold change.

**Table 3 ijms-23-06421-t003:** Target genes knocked out in RAW 264.7 cells using the CRISPR-Cas9 tool.

Gene	gRNA Sequence	Knockout Sequence
*Aif1*	GTCCAAACTTGAAGCCTTCA TGCTGTATTTGGGATCATCG	Not detected
*Calcrl*	CCCAGGTCCTATTGCAGTAA	GCTGACCC---//---CTGGAATGAC (56-base pair deletion)
*Gsta3*	GGAGCCTATCCGGTGGCTCT TCCTTCATTACTTTGATGGC	Not detected
*Ifit2*	ATCAGAAGTCTGGTCACCTG	Not detected
*Marco*	CAGCACCCAATCTGAGAGAA	GGGGACCT--/TCTC/TGCATGGCA (2-base pair deletion or addition)
*Tmem26*	TTCCAATTACACGAATTAAA	GCAGCACC---//---CACAGAACT (16-base pair deletion)
*Ube3a*	GTCCAAACTTGAAGCCTTCA ATATACAAGTGCATTCAGGA	AACTGCCTT-CTGAATGCACTTGT (1-base pair deletion)

**Table 4 ijms-23-06421-t004:** Primers for qRT-PCR.

Gene	Sequence
*Csf1r*	CTTCACTCCGGTGGTGGTGG and GCGCACCTGGTACTTCGGCT
*Itgb3*	ACAGAGCGTGTCCCGTAATC and GTCTTCCATCCAGGGCAATA
*Traf6*	AAAGCGAGAGATTCTTTCCCTG and ACTGGGGACAATTCACTAGAGC
*Nfatc1*	CCCGTCACATTCTGGTCCAT and CAAGTAACCGTGTAGCTCCACAA
*Rela*	GCCCAGACCGCAGTATCC and GTCCCGCACTGTCACCTG
*c-Src*	CCAGGCTGAGGAGTGGTACT and GAGCTTGCGGATCTTGTAGT
*Atf3*	GCTGCTGCCAAGTGTCGAAA and TACATGCTCAACCTGCACCG
*Fosb*	GATCGCCGAGCTGCAAAAAG and CCTTAGCGGATGTTGACCCTGG
*DC-stamp*	GGGAGTCCTGCACCATATGG and AGGCCAGTGCTGACTAGGATGA
*OC-stamp*	CAGAGTGACCACCTGAACAAACA and TGCCTGAGGTCCCTGTGACT
*Calcr*	CCTTCCAGAGGAGAAGAAACC and GGAGATTCCGCCTTTTCAC
*c-Fos*	CCAAGCGGAGACAGATCAACTT and TCCAGTTTTTCCTTCTCTTTCAGCAGA
*Oscar*	TGGCGGTTTGCACTCTTCA and GGAAGAACTCAGCCAGCTCAA
*Aif1*	TGATGAGGATCTGCCGTCCAAACT and TCTCCAGCATTCGCTTCAAGGACA
*Calcrl*	CAAGATCATGACGGCTCAATA and CGTCATTCCAGCATAGCCAT
*Gsta3*	AACCGTTACTTTCCTGCCTTTG and GCCCTGCTCAGCCTATTGC
*Ifit2*	AAATGTCATGGGTACTGGAGTT and ATGGCAATTATCAAGTTTGTGG
*Marco*	AGGAAGACTTCTTGGGCAGC and GAGCAGGATCAGGTGGATGG
*Tmem26*	GCACCATCACTAGAGACCAAC and ACAAGAATGCCAGAGACCAG
*Ube3a*	CAGCCTAGTTCAAGGACAGCAG and TCCACATACAACTGCTTCTTCAAG

## Data Availability

Available upon request.

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
