# Peer review of "Identification of Novel Genes for Cell Fusion during Osteoclast Formation"

_ijms, 2022, doi:10.3390/ijms23126421_

Round 1
Reviewer 1 Report
The manuscript entitled “Identification of novel genes for cell fusion during osteoclast formation”, by Eunjin Cho et al., allowed authors to identify novel markers of osteoclast differentiation that are, more specifically, involved in the transistion from mononuclear to multi-nucleated cells of this lineage. This issue was accomplished using an RNA - seq approach, aimed to identify genes exhibing a differential expression pattern between the two investigated cell contexts, followed by their functional validation carried out with a CrispR – Cas9 based procedure.
There is no doubt that the authors decided to study a relevant aspect of osteoclastogenesis, using the best and more advanced technologies that are available for this purpose and performing an impressive number of experiments that allowed them to obtain very interesting results.
Unfortunately, in several parts of the manuscript, methodological aspects are not properly explained or not convincing, the biological rationale of experiments is often confused and the data obtained are not clearly presented. Perhaps it would be convenient for authors to reduce the amount of data contained in the manuscript, trying to improve their focus, rigour and clarity. English style should also be improved in several parts of the manuscript.
I will try to list below some of the problems that I observed throughout the manuscript.
- To begin, how did authors obtain murine Bone Marrow Macrophages (BMM)? Are they sure they didn’t make confusion with total Bone Marrow cells and actually started their culture experiments using this cell sample in place of BMM? If this happened, it is not as problem but the important is that authors clarify the real biological nature of their starting material. “Rough dissociation” (as authors define it) appears to me an inaccurate and questionable procedure to separate SNC form MNC. What was the purity of the three analyzed cell populations (BMM, SNC, MNC)? Were these cell populations highly purified or simply enriched? What methods did authors adopt to assess this aspect? Did they employ morphological and / or flow cytometry analysis? Authors should answer these questions indicating the purity percentages of BMM, SNC and MNC samples and the procedures used to obtain the corresponding values.
- In Figure 1, panel C it is not clear why authors created the Venn diagram, and performed the relative analysis, using the lists of differentially expressed genes deriving by the three comparisons (SNC–MNC, BBM–SNC, BBM–MNC) rather than using “crude” lists of genes expressed by the three considered cell populations (BBM, SNC, MNC) more suitable, in my opinion, to investigate the phenotypic similarities existing among them.
- In the second part of Paragraph 3.2, I think it would be more correct to refer to the considered cell populations calling them “adherent cells” and “non adherent cells”, or as alternative for the latter, “suspension cells” as the same authors did sometimes, avoiding to use definitions that, to my knowledge, have been never used before in the scientific literature such as “suspended” or “suspending” cells.
- Always on this part of the manuscript, whoever has dealt with the monocyte / macrophage system, knows that all cells belonging to this lineage, including macrophages and osteoclasts, efficiently adhere to the culture flask within few hours from seeding. Therefore, it is surprising that authors observe a subpopulation of cells not adhering to the substrate at longer times. Are they sure of the real nature of these cells? Did authors perform any morphological and / or flow cytometry analysis to verify this issue?
- Again, the experiments thereby described, the results of which are presented in Figure 4, were not clearly explained, although probably with the exception of panel A. Considering that half samples in this figure are represented by non adherent cells, it is not clear how these cells (panel A, upper image; panel B, upper right image; panel C upper right image; panel D) underwent the indicated cytochemical and fluorescence analysis. Were these cells cytocentrifuged? On the contrary case it should be concluded that they are actually adherent cells, contradicting the author claim. In panel B, authors did not precisely explain which of the two cell populations was labelled with one fluorochrome or the other. If I understood correctly, by this experiment, authors tried to demonstrate that MNC can derive by SNC. Reasonably, and as expected, the opposite finding is not claimed by authors. Based on this premise it is not clear why both cell populations exhibit both fluorescence signals (green and red). If the assumption of authors was correct, this behaviour would be expected only for non adherent cells. In this regard it is also worth underlining that the interpretation of these results are also greatly hampered by the circumstance that multi-nucleated osteoclasts can be hardly visualized in the images presented thereby. Therefore authors should provide higher quality images obtained at higher magnification values. The experiment presented in panel C, at least to me, appears incomprehensible.
- The presence of Ki-67 positivity in panel D is more than surprising, because monocytes, macrophages and osteoclasts are all post-mitotic, quiescent and therefore non proliferating cells. Are authors sure of the nature of these cells? How was this antigen labelled? What negative and positive control were carried out to verify the reliability of antibody labelling? Did authors consider the possibility that Ki-67 positive cells are actually represented by precursor blast cells?
- In a generally accepted vision of osteoclatogenesis, macrophages give rise to mononuclear osteoclasts, in turn generating multinucleated osteoclasts. Authors appeared to share this principle in several parts of their manuscript. Therefore, at the beginning of paragraph 3.4, it is not clear why they tried to identify genes that are specifically associated with the differentiation transition from SNC to MNC considering the BMM–SNC and BMM–MNC comparisons rather than focusing their attention on the SNC–MNC comparison. Are they exploring the possibility that in addition to a typical origin from SNC, MNC can also directly derive from BMM? Authors should be more clear on this point.
- In Figure 8, panel C, at day 4 of stimulation with Rankl, the difference of Calcr protein expression, observed between the Wild-type and Knock-out samples, is very weak and needs to be quantified through a densitometric analysis.
- At the beginning of Discussion authors state that, among other different cell types, BMM can give rise to myeloid blasts. Similarly, in the middle of Discussion, authors say that BMM active are able to proliferate actively. These assertions are both absolutely wrong and, again, I’m afraid that authors make confusion among different hematopoietic cell types. If BMM, as authors declare in the Introduction, stay for Bone Marrow Macrophages, these cells cannot proliferate and cannot differentiate to generate myeloid blast cells. On the other hand, these assertions can become true if BMM are actually total Bone marrow cells or a fraction of Bone Marrow tissue represented by myeloid precursors (synonymous of myeloid blasts). Therefore, authors should explain precisely how they obtained what they call BMM.
- Authors should explain, in Materials and Methods and / or elsewhere, what one asterisk (*) and two asterisks (**) stay for in terms of p value and statistical or highly statistical significance.
Reviewer 2 Report
In this manuscript, the authors performed RNAseq with single nucleated cells (SNCs) and multinucleated cells (MNCs) to identify the fusion-specific genes. They found 7 genes increased in MNCs whereas this was decreased in SNCs compared with BMMs. They use CRISPR-Cas9 genome editing tool to examine their function during osteoclastogenesis. Based on their data, they concluded Calcrl, Marco, and Ube3a are novel determinants in osteoclastogenesis, especially cell fusion. Although lots of work were done, some important questions must be addressed before acceptance for publishing.
Main concerns:
- The authors claimed that SNCs are osteoclast committed cells, why expression level of the 7 genes they identified are lower in SNCs compared with macrophages?
- In the whole paper, the authors used cell dissociation buffer to separate SNCs (detached) from MNCs (adherent, un-lifted by dissociation buffer) on Day 4 for gene array and Day 2 for cell culture. While it is easy to lift SNCs, could the authors explain how could they lift the MNCs for co-culture?
- Cell density is a critical factor affecting in vitro osteoclast differentiation. In Fig 4C, the authors found that more fused cells were detected in the coculture with suspended cells than with adherent cells, one possibility is adherent cell density is too high for osteoclast fusion in adherent cell coculture.
- Calcrl is one of the 7 genes they found that are high expression in MNCs compare with macrophages. But the data in fig7a shows it is only high expressed on D2 with RANKL, at that time almost no MNC formed. Also in this fig, its expression on Day4 and day5 is much lower than macrophages (D0). In 7B, Calcrl immunostaining shows not only MNCs, but also the surrounded SNCs have high Calcrl expression, which does not fit the RNAseq data that its expression decreased in SNCs compared with macrophages.
Author Response
We thank the reviewers for their helpful comments. We have thoroughly revised the manuscript according to the comments, and hope that the revised manuscript is suitable for publication in Bone. Revised texts are marked by highlights.
- The authors claimed that SNCs are osteoclast committed cells, why expression level of the 7 genes they identified are lower in SNCs compared with macrophages?
Thank you for your question. This is why we have suggested that they are fusion-specific genes. It is possible that SNCs are slightly delayed cells or not sufficient in becoming MNCs, and we have indicated in Figure 4 that SNCs could fuse in a new environment. Another possibility is that SNCs maintain in vivo heterogeneity for fusion processes, suggesting that fusion occurred between SNCs and MNCs, as mentioned in the Discussion.
- In the whole paper, the authors used cell dissociation buffer to separate SNCs (detached) from MNCs (adherent, un-lifted by dissociation buffer) on Day 4 for gene array and Day 2 for cell culture. While it is easy to lift SNCs, could the authors explain how could they lift the MNCs for co-culture?
Thank you for your concerns. On day 2 of co-culture, we scraped the attached cells to detach them. We have explained this in more detail in the Materials and Methods section 4.4. as follows: “To assess fusion, a fusion assay was performed via cell labeling [60]. Briefly, the BMMs were cultured for 2 days under osteoclastogenic conditions. Cells in suspension subjected to RANKL stimulation were collected using cytocentrifugation. Adherent cells were directly labeled on the plate for 30 min and were then scraped off. Cells were labeled with Cell Tracker Green CMFDA or Cell Tracker CM-DiI (Thermo Scientific, Waltham, MA, USA). Differently labeled cells were mixed and co-cultured, or the DiI-labeled cells were co-cultured with non-labeled adherent cells for 2–3 days in an osteogenic medium.” (Page 16; Lines 447–484).
Additionally, although MNCs on days 4–5 were hard to detach, the cells on day 2, containing 2–3 nuclei, were easy to detach. We assume that fully differentiated cells are strongly attached, whereas immature osteoclasts are less adherent.
- Cell density is a critical factor affecting in vitro osteoclast differentiation. In Fig 4C, the authors found that more fused cells were detected in the coculture with suspended cells than with adherent cells, one possibility is adherent cell density is too high for osteoclast fusion in adherent cell coculture.
Thank you for this suggestion. However, as seen in Figure 4B and 4D, we tested cell fusion at a high cell density. We do not believe that the cell density shown in Figure C is too high. Moreover, we counted cells depending on the capacity of the culture plate, and this experiment was performed with fewer cells than the number used under normal differentiation conditions owing to the quality of the microscope images.
- Calcrl is one of the 7 genes they found that are high expression in MNCs compare with macrophages. But the data in fig7a shows it is only high expressed on D2 with RANKL, at that time almost no MNC formed. Also in this fig, its expression on Day4 and day5 is much lower than macrophages (D0). In 7B, Calcrl immunostaining shows not only MNCs, but also the surrounded SNCs have high Calcrl expression, which does not fit the RNAseq data that its expression decreased in SNCs compared with macrophages.
Thank you for the concerns; we apologize for the ambiguously-worded text. In Figure 7A, we tested Calcrl mRNA levels in whole cell lysates, not separately in SNCs and MNCs. Additionally, when we tested mRNA levels of our fusion-specific genes in whole cell lysates, the expression pattern was similar to that in SNCs, and the levels were suppressed in mature osteoclasts on day 4. We have mentioned this in the Discussion. (Page 15; Lines 414–419).
In Figure 7B, we assume that the dotted-like pattern in the MNCs represents the actual Calcrl expression because Calcrl is a membrane protein. We have indicated this with arrows in the figure.

Round 2
Reviewer 1 Report
I think that author reply is satisfactory and, therefore, the last version of the manuscript can be published.
Author Response
Thank you.
Reviewer 2 Report
The authors did not fully address my concerns.
1) They did not explain well why Calcrl only high express on Day2 with RANKL. This is totally against their conclusion that it is important for cell fusion and only high expressed on MNCs. I am more confused by their discussion on line 414-419.
2) The authors must improve the quality of Fig 7B. For me, the Calcrl immunohistochemical staining was not successful, they were non-specific.
3) They should show all data they got, not “data not shown”
Round 3
Reviewer 2 Report
see attached file.

Round 4
Reviewer 2 Report
It is OK to accept this manuscript for publication.
This manuscript is a resubmission of an earlier submission. The following is a list of the peer review reports and author responses from that submission.
Round 1
Reviewer 1 Report
In their study entitled “Identification of novel genes for cell fusion during osteoclast formation”, Eunjin Cho and colleagues used an RNAseq approach to try and identify regulators of the cell-cell fusion process that occurs during osteoclast differentiation.
The molecular mechanisms involved in osteoclast precursor cell fusion remains poorly understood and very few regulators have been identified so far, such as CD47, DC-STAMP, ATP6v0d2 and Adrm1. Identify novel regulatory genes is of interest.
The strategy here to identify gene involved in fusion was to perform RNAseq and compare bone marrow macrophages (BMM) used as osteoblast precursors, and, after 4 days of culture with M-CSF and RANKL, the multinucleated wells (MGC), which are considered differentiated osteoclasts, and single nucleated cells (SNC) that remained in the culture and were collected by treatment with a cell dissociation agent.
The present study proposes three new regulators of osteoclast precursor cell fusion: Calcrl, Marco, and Ube3. Nevertheless, a major weakness here is that there is no clear evidence that these genes are indeed involved in the fusion process else than multinucleated cells do not form. Regarding the KO of Calcrl for instance, the expression of osteoclast-characteristic genes is impaired such as Oscar and Acp5, which expression is independent of cell fusion as reported for instance for ATP6v0d2 KO osteoclasts. Calcrl rather appears as a regulator of the RANKL-induced transcriptional program induced by RANKL. The expression of osteoclast markers was not assessed for Marco and Ube3 KO.
Also, most of the RNAseq results tested were not confirmed by Q-PCR and the number of samples analyzed by RNAseq is too low to perform statistical analysis of differential gene expression. How do the genes known to be involved in osteoclast fusion behave in the RNAseq and in Q-PCR?
Specific comments:
- RNAseq study:
There is no information regarding the quality of the RNAseq and the number of genes identified, a Venn diagram of each cell category would allow comparing the 3 types of samples. The PCA in Figure 1B actually shows that the 2 SNC and the 3 MGC samples do not segregate.
There are only 2 samples for SNC and 3 samples for BMM and MGC, therefore it is not possible to perform a statistical analysis, all the more as multiple comparisons are made. How is it possible to calculate a q-value?
All genes were considered that had FPKM>1 in at least one of the samples. This is extremely low; usually, for differential gene expression studies, only FKPM>10 are considered. No threshold is mentioned to define differentially expressed genes, which is very unusual; does this mean that the only criterion was that the q-value was over 0.05, whatever the fold change?
The conclusions drawn from the RNAseq data can be questioned in fact, given the numerous discrepancies with the Q-PCR results. For instance in figure 6, among the seven genes picked by the authors as relevant and highlighted in Table 1 (“Interestingly, only seven genes revealed opposite expression patterns that were significantly downregulated in SNCs compared to BMMs, whereas these were upregulated in MGCs compared to BMMs”), not less than 4 show a different behavior in the Q-PRC validation in Figure 6B, as compared to RNAseq results shown in table 1: Calcrl (up in SNC), Gsta3 (down in MGC), Tmem26 (up in SNC) and Ube3a (down in MGC). The graph in Figure 6A shown more spots that fulfil the opposite expression pattern between SNC ad MGC, why weren’t selected?
It is extremely surprising that the expression of osteoclast characteristic receptor RANK (Tnfrsf11a), which governs osteoclast differentiation and is strongly induced by RANKL, is not expressed at higher levels in RANKL treated cells (NSC and MGC) as compared to BMMs.
The gene ontology analyses presented in figures 2B and 5B, C are highly surprising: the TCA cycle and the mitochondrial metabolism-related genes were documented in several studies to be strongly induced during osteoclast differentiation (An et al, 2014; Lemma et al, 2016; Guerit et al, 2020…). None of these studies are cited actually and no comparison with previous transcriptomic studies is made.
The gene ontology analyses should show the fold enrichment as compared to what is expected, not the p-values and the crude counts. The definition of the reference transcriptome is also missing: is it all the transcripts found in this study or the whole mouse transcriptome?
The analysis of the RNAseq data is clearly missing the direct transcriptome-wide comparison between SNC and MGC, to highlight the actual differences between these two types of RANKL treated cells, as they were used to identify genes involved in fusion. As such, the analysis is split between Figures 1, 2, 3, and 5 making it difficult to figure out to what extent the SNC and MGC, which do not segregate in the PCA of figure 1, have a distinct transcriptional program.
- Q-PCR analyses: figures 3B, 6B and 7D.
Except for 7D, there is no mention of the significance of the differences.
About Figure 3, the authors claim “Some genes, such as NFATc1, c-Src, OC-STAMP, and calcitonin receptor (Calcr) had higher expression in SNCs than MGCs, correlating with the RNA-seq results”. This may be true for the Q-PCR on Nfatc1 but it does not seem to be the case for c-Src, OC-STAMP, and Calcr in 3B: expression in SCN is not different (OC-STAMP) or very marginally higher (c-Src and Calcr). The significance of these marginal differences must be tested. It is not really possible to compare with the RNAseq results, as numerical data are missing: only a heatmap is shown with apparently the same color in SNC and MGC. In 3A fact, the scale of the RNAseq heat map is very puzzling: “Log2(FKPM)”, only ranging from 0 to 10. It would be much more informative to have the actual FKPM numbers. Also, RNAseq data should be provided in 3A for all the genes analyzed by QPCR in 3B, Atf3 and Ctsk are missing. The authors should comment on the fact that there is no induction of Tnfrsf11a in SNC and MGC as compared to BMM while the expression of RANK receptor is characteristic of RANKL-induced osteoclast differentiation.
About figure 6, as stated above the QPCR data do not confirm the RNAseq profile, which motivated the selection of the 7 genes tested. What is the culture time point used to prepare the RNA to perform the Q-PCR in 6B?
Figure 7D shows that the Calcrl KO cells actually do not respond to RANKL: there is no induction of Oscar and Acp5, so it is not possible to conclude that the defect in multinucleation arises from a fusion problem, it is a lack of response to the cytokine.
- Osteoclast data:
Cell density information is missing in all the experiments.
The legend of Figure AB mentions “These adherent and suspended cells were stained with CMFDA and DiI, respectively, and then co-cultured for 3 days in RANKL-containing medium.” In fact, the experiment mixed either adherent cells labeled with CMFDA and adherent cells labeled with DiI (top panels) or non-adherent cells labeled with CMFDA and non-adherent cells labeled with DiI (bottom panels). It would be more informative in 4C to label the adherent cells with CMFDA to see the fusion events with the added DiI labeled cells. How were adherent cells labeled with DiI collected to make the co-culture? All the experiments in Figure 4 need quantification.
Figure 6D: what does “Fit area” mean?
Figure 7: how does the expression of CALCRL protein in SNC and MGC correlate with the RNAseq and Q-PCR data?
Reviewer 2 Report
By doing RNA-Seq, Cho et al. revealed differentially expressed genes between pre-fused single nuclear (SNCs) and fused multinucleated osteoclasts (MNCs) derived from mouse primary bone marrow monocytes in vitro. Compared to bone marrow derived macrophages, seven genes were found upregulated in MNCs and downregulated in SNCs. Among them, three genes were considered as osteoclastogenesis-associated genes in RAW264.7 cells used as osteoclast precursors in this study.
Comments:
- Results obtained by the authors were misinterpreted in the title, and elsewhere in the manuscript. Although RAW264.7 cells in absence of Calcrl, Marco, and Ube3a were unable to turn to multinucleated osteoclasts, this does not necessarily mean they are genes for cell fusion, especially when genes are highly expressed in cells that already fused, plus cell fusion is a transient event in osteoclast differentiation.
- Typos such as 7days (7 days) in Line 97; “western blotting” (Western blotting) in the entire manuscript; Line 152 “Student’s t-tests” (Student’s t-test); “pre-osteoclast” and “preosteoclast”. (Should be consistent).
- In order not to mix up with multinucleated giant macrophages, better replace MGCs with MNCs or MNOCs as the abbreviation for “multinucleated osteoclasts”.
- Line 123-125, sequence of the experimental procedure is incorrect. cDNA synthesizes first, and then library preparation.
- Line 161 to 163, if SNCs are cells in the middle stage (osteoclast-committed) of osteoclastogenesis, the authors should discuss why those upregulated genes in the multinucleated cells were down regulated in SNCs compared to macrophages (osteoclast precursor cells beyond primary bone marrow monocytes).
- 7C, resolution of Carcrl immunoblot (day 3 and 4) is low. The authors will have to redo it. “-“ and “+” in the figure need to be specified in the figure legend.
Reviewer 3 Report
Summary:
In this study, the authors used compared the transcript mRNA expression level and pattern between BMMS, SNCs and MGCs groups using RNAseq. From the analysis, an interesting group of genes that differ between MGCs and SNCs against BMMs. In vitro validation using CRISPR-Cas9 KO, TRAP staining and resorption assay confirmed the importance of the identified genes. Overall, this manuscript provides a rather complete research design. However, some critical details and discussions are missing. Minor changes are needed before publication.
Comments:
- It is not clear whether all the cells lysed at the same time. Description is only provided for the MGCs group.
- Figure 2B, discussion regarding the mitochondria expression upregulation in SNC vs. MGC is needed.
- Please specify the variant and control in comparisons such as SNC-MGC in figure 2A. Based on the context, the upregulated DEGs in SNC-MGC comparison are referring to the genes were upregulated in MGC group compared to the SNC group. This is very confusing as the SNC-MGC symbol might suggest the other way.
- More detailed information should be provided regarding the RNA sequencing depth and the total number of genes that were matched.
Specific comments:
- Page 2, line 68, italicize “in vitro” and please check throughout the manuscript.
- Please provide the full name or explanation when fist introducing a less frequently used terminology such as q-value, rough dissociation.
- Figure 1, please keep the color scheme between panels consistent and provide a legend for Venn diagram (not the interaction groups but the original groups).
- Figure 3, Figure 5 B and C, please make sure the figure border and cell border are consistent.
- Please label or add in the caption of figure 2B with “comparison between SNC and MGC”.
- Please reconsider 3.2 subtitle.
- Figure 3A could be confusing since the metrics is switched from FC to FPKM. Please indicate that in the context and caption.